# Feeding Problems Assessment Tools in Children: A Scoping Review

**DOI:** 10.3390/children12010037

**Published:** 2024-12-28

**Authors:** Suci Destriatania, Judhiastuty Februhartanty, Fariz Nurwidya, Rini Sekartini

**Affiliations:** 1Department of Nutrition, Faculty of Medicine, Universitas Indonesia, Dr Cipto Mangunkusumo General Hospital, Jakarta 10430, Indonesia; suci_destriatania@fkm.unsri.ac.id; 2Department of Nutrition, Faculty of Public Health, Universitas Sriwijaya, Kampus Unsri Indralaya, Ogan Ilir 30662, Indonesia; 3Southeast Asian Ministers of Education Organization Regional Centre for Food and Nutrition (SEAMEO RECFON) PKGR Universitas Indonesia, Jakarta 13120, Indonesia; 4Department of Pulmonology and Respiratory Medicine, Faculty of Medicine, Universitas Indonesia, Jakarta 10430, Indonesia; fariz.nurwidya@ui.ac.id; 5Department of Child Health, Faculty of Medicine, Universitas Indonesia, Dr. Cipto Mangunkusumo General Hospital, Jakarta 10430, Indonesia; rini.sekartini@ui.ac.id

**Keywords:** child feeding, feeding problems, feeding problems assessment tool

## Abstract

‘Feeding problems’ is a term used to describe problems that may present typically in children. Problems with feeding during infancy can result in significant negative consequences for a child’s nutrition, growth, and brain development. This scoping review aims to map current research, provide summary of the available feeding problem assessment tools for children, and review current implications and the gaps between tools, providing information that academics, practitioners, and parents may find useful. Three electronic databases (PubMed, Science Direct, and ProQuest) were searched using terms related to feeding problem assessment tools in children, which included, but were not limited to, “feeding difficult*”, “eating problem”, “eating difficult*”, “tool”, “child*”, and “pediatric”. The following limits were implemented on the search: English language, age limit (<18 years old) and publication period (last 10 years). Data management and analysis carried out manually through discussion with the team. Authors 1 and 2 screened titles and abstracts, then full texts were discussed with the full team to identify articles that met inclusion and exclusion criteria. Data were charted into a matrix table based on these categories: author, year, population, assessment tools, usage and aspects. Thematic analysis was carried out to summarize the characteristics of the studies. There were 47 papers included in the study and analysis, in which 23 assessment tools were found. Pedi-EAT was the most frequent assessment tool used in the studies, with nine papers covering this feeding problem assessment tool. MCH–FS came in second for its chosen tool quantifying children’s feeding problems, with a total of seven papers covering this tool, along with BPFAS with seven papers. In this review, 23 assessment tools were validated and tested for reliability. Pedi-EAT, MCH-FS and BPFAS were commonly used instruments. However, it is clear that no single instrument covers comprehensively all aspects of feeding problems in children. In addition, usage of the tools and wide age range indicate that further research is needed to fill the gaps.

## 1. Introduction

Feeding problems in early childhood are common and raise significant concern for both parents and pediatricians globally. Estimates of prevalence of feeding problems in typically developing children vary widely in the literature, between approximately 25 and 45% [1,2]. This is due to differences in assessment methods and inconsistencies in definitions. The various terms used to describe feeding-related issues seem to be employed differently. For instance, the term “feeding problems” and “feeding difficulties” have become confusing because they are often used together in the same study. Kerzner described ‘feeding difficulty’ as an umbrella term that broadly indicates the presence of various forms of feeding problem [3]. ‘Feeding problems’ is one of the terms used to describe a variety of problems that may present as, but not limited to, a lack of age-appropriate feeding skills, inappropriate eating habits, disruptive mealtime behavior, family conflicts due to feeding, and an unpleasant atmosphere during meals [3]. During the first two years of life, children’s eating habits are strongly influenced by parental feeding practices. This is because their ability to eat and their nutritional needs change as they grow and learn new skills [4]. Effective feeding requires a parent or caregiver who trusts and relies on the child’s cues regarding timing, portion size, preferences, pace and ability to eat. Problems with feeding during infancy can lead to significant negative consequences on a child’s nutrition, growth, and brain development [5,6].

Currently, there are no universally accepted tools to diagnose feeding problems in children, not to describe healthy feeding. The assessment tools available are very specialized, typically tailored to a specific medical condition. The complicated and varied issues that underly feeding problems are some of the potential causes of scarcity in feeding problems assessment tools for children. However, advancement regarding these feeding problems assessment tools has developed in recent years, so a number of standardized psychometric tools have been developed increasingly over the past 20 years to assess child feeding problems. Existing measures include the Children’s Eating Behavior Inventory (CEBI), Behavioral Pediatric Feeding Assessment Scale (BPFAS), Mealtime Behavior Questionnaire (MBQ), The Pediatric Eating Assessment Tool (Pedi-EAT), The Screening Tool of Feeding Problems applied to Children (STEP-CHILD), The Montreal Children’s Hospital Feeding Scale (MCH-FS), and many more that will be covered in this study [7].

Due to the importance of the right usage of feeding problems assessment tools for children with a variety of backgrounds or conditions, further analysis of each tool is needed. That is why this scoping review aims to map current research, provide a summary of the available feeding problems assessment tools for children, review current implications and examine gaps between tools. This will provide information that both academics, practitioners, and parents may find useful.

## 2. Materials and Methods

This scoping review adapted the method from Hielscher et al. [8]. The original protocol, which was originally used to complement feeding in Down Syndrome, was modified with enhancements from Deandra et al. [9]. However, the protocol of the scoping review was not pre-registered. This study then reported in line with the Preferred Reporting Items for Systematic Reviews and Meta Analysis (PRISMA) extension for scoping reviews [10]. The stages taken in this study were: (1) identifying research questions, (2) identifying assessment tools (3) study selection (4) charting the data, and lastly (5) combining, summarizing, and reporting the data.

### 2.1. Research Question

Examination of the available data raised key questions regarding the assessment tools used to identify feeding difficulties in children:What are the tools used to identify feeding problems in children? How are these tools implemented?What are the implications of each tool used to identify feeding problems in children?

### 2.2. Search Strategy

A literature search was conducted using electronic databases to identify relevant texts, including PubMed, Science Direct, and ProQuest. The search parameters included combinations with function “AND” of key terms relating to the assessment of feeding problems in children: “feeding problem”, “assessment”, “instrument”, and “infant”. Variations of the keywords were combined with the “OR” operation to maximize results: “feeding difficult*”, “eating problem”, “eating difficult*”, “tool”, “child*”, and “pediatric”. An asterisk (*) was used to gather different wordings of the same meaning, such as ‘difficulty’ or ‘difficulties’. This review only considered articles in English and published in the 10 years between August 2014 and August 2024.

### 2.3. Study Selection

Inclusion criteria for this scoping review included studies related to any aspect of feeding problems assessment tools (development, implementation, evaluation) for children who started complementary feeding or eating (maximum 18 years old) in any country using any type of methodology. This review included implementation and/or implication of feeding problems assessment tools, problems, regulations, guidelines, and recommendations applicable for children. Exclusion criteria consisted of assessment tools for adults (above 18 years old), assessments that were not questionnaire based, articles in non-English language, breastfeeding/bottle feeding articles, unavailable abstract and manuscripts, and publications before August 2014 or after August 2024.

Feeding problems have been categorized and/or classified using a variety of terms, including multiple potential underlying causes which make it challenging to develop appropriate tools. Therefore, when surveying and collecting the literature, we aimed to be more inclusive in order to capture a variety of existing tools. The existing measurement tools are either generalized, looking at the overall eating habits of the pediatric population over a broad age range (<18 years old), or very specialized measurement tools, like those for children with specific medical conditions.

One author (SD) conducted the searches for relevant titles and abstract from the databases using the search terms. Article titles and abstracts were then manually screened to identify relevant articles. Discussions between first and second author screened the selected articles. Full text articles were retrieved and reviewed through discussion with the full team in order to make a final decision regarding inclusion. Authors 1 and 2 decided to exclude 22 articles related to breastfeeding/bottle feeding assessment. This was because the articles were not relevant to the inclusion criteria of the study.

We developed a matrix table as a charting tool to map study characteristics into categories. All team members discussed the charted data based on the research questions and entered the data into a charting tool. Group discussions regarding the characterization and organization of the charted data allowed for refinement of the categories and the development of additional categories. At each iteration of the data charting process, we took similar steps to refine and reach consensus regarding data charting and the development of final categories, which included the following for each article:Year—to define the recency of the study following the inclusion criteria for the years 2014–2024.Assessment tools, to provide the name of these tools.Population, to identify the study participants or cases fort whom the measurement tools were developed or implemented. This information can help identify the age groups of children who have been either frequently studied or understudied in research for the development and implementation of feeding assessment tools.Usage, to determine the purpose of the measurement tools.Aspects, to describe issues such as (a) whether the tool was designed for clinicians/medical professional or for mothers/caregivers and (b) other aspects of the feeding problems, as items developed or measured in the study.

This data organizing process allowed for systematic organization of the findings and improved consistency in interpreting the data. To ensure accuracy, data charting was checked independently by each member of the team. Finally, papers included in this study were used to answer the study’s research questions regarding feeding problem assessment tools in children and to cover the following themes: availability, implementation, and implications of feeding problems assessment tools in children.

## 3. Results

The article selection process is summarized in Figure 1. The systematic searches identified a total of 822 articles. Duplicates were then removed and a screening process based on title and abstract were carried out. This process yielded 69 articles. Although these 69 articles fulfilled the inclusion criteria, 22 were removed, either for evaluating breastfeeding or bottle feeding. Therefore, 47 papers were finally included in the analysis to answer the study’s research questions regarding feeding problem assessment tools in children and to cover the following themes: availability, implementation, and implications of feeding problems assessment tools in children.

### 3.1. Availability and Implementation of Feeding Problems Tools in Children

From the data in Table 1, it can be seen that there are no generally accepted tools for feeding problems in children. From three databases, 47 papers were collected, and 23 tools were found. Pedi-EAT was the most frequent assessment tool used in the studies, with nine papers covering this feeding problem assessment tool, which serves to identify symptoms of problematic feeding in children. MCH-FS came in second for its chosen tool used to quantify children’s feeding problems, with a total of seven papers, along with BPFAS with seven papers. Different papers used different terms to describe the usage of assessment tools. This could be due to the fact that all three instruments incorporated both children from clinical samples and healthy children to evaluate feeding difficulties in their study population. For example, Pedi-EAT was described as “assessment of symptoms of problematic feeding in infants and young children” [11] but other paper described it as an evaluation of dysphagia symptom severity in children with autism [12]. This difference could lead to misuse of tools, or even misdiagnosis and mistreatment in the long run.

Furthermore, other assessment tools assessed in this review were used to assess specific aspects of feeding or feeding problems related to children’s medical conditions, so examined specific aspects or conditions, and therefore are not as commonly used as the three assessment tools previously mentioned. For example, the Brief Autism Mealtime Behavior Inventory (BAMBI), Feeding Interaction Scale (FIS), Screening Tool of Feeding Problems Applied to Children (STEP-CHILD), and Autism Eating Questionnaire (AUT EAT) were developed specifically for children with autism, while the Diabetes Eating Problem Survey Revised (DEPS-R) is used for early detection of eating behavior issues in diabetic children. The Eating and Drinking Ability Classification System (EDACS) is employed to evaluate eating and drinking efficiency in children with cerebral palsy. The Feeding Handicap Index for Children (FHI-C) was designed to assess feeding problems in children with developmental disabilities. Specific feeding issues, such as food neophobia, emotional eating status and parents’ feeding practices, are assessed by the Food Neophobia Scale for Children (FNSC), the Child Food Neophobia Scale (CFNS), Emotional Eating Scale for use in Children and Adolescents (EES-C) and the Stanford Feeding Questionnaire (SFQ), respectively.

Pedi-EAT, MCH-FS and BPFAS incorporated aspects of feeding behavior and the feeding capacity of children when assessing feeding difficulties. However, MCH-FS and BPFAS added aspects such as parental perception, strategies and interaction at mealtimes. Most items in the reviewed instruments were created via expert discussions, theoretical frameworks, previous instruments, and literature reviews. Interestingly, Pedi-EAT was the only example that took parent perspectives into account when generating its items. However, a limitation of Pedi-EAT is the lack of sensitivity in its scoring system to the child’s age when examining feeding problem symptoms, particularly in young children. MCH-FS is designed for quick assessment of feeding problems in clinician’s offices, while Pedi-EAT and BPFAS were parent-reported assessment tools.

Content validity has been established for Pedi-EAT, MCH-FS, and BPFAS. Pedi-EAT demonstrated strong content validity, with high relevance and clarity scores. A well-documented content validation process was provided in this prominent study. Its items were derived from multiple information sources and validated by a team of multidisciplinary experts, as well as parents of children with feeding difficulties. However, the content validity of the MCH-FS and BPFAS has been confirmed only by experts (e.g., psychologists), but details on the validation process were not provided.

Regarding structural validity, factor analysis has been conducted for Pedi-EAT, MCH-FS, and BPFAS. Among these, only BPFAS has been confirmed to have a good model fit through confirmatory factor analysis. Meanwhile, the items in the Pedi-EAT and MCH-FS were developed using exploratory factor analysis, revealing moderate variance between items and constructs.

Internal consistency has been established for all the reviewed instruments, with Pedi-EAT ranging from good to excellent, BPFAS ranging from acceptable to excellent, and MCH-FS ranging from unacceptable to good.

### 3.2. Implications of Feeding Problems Tool in Children

The questionnaire created for children with Autism Spectrum Disorder (ASD) was more thoroughly and comprehensively tested than those for Typically Developing Children (TDC). The objective of these screening tools should not be to screen children’s feeding problems when there are already obvious symptoms and feeding disorders, but to observe transient minor feeding concerns. Therefore, caregivers could make changes regarding mealtimes and behavior to prevent feeding disorders in children. Unfortunately, not many tools were designed for the latter purpose. However, the Infant and Child Feeding Questionnaire (ICFQ) tool highlights this importance, as it described how earlier identification and treatment of pediatric feeding disorders could act in preventing development of comorbid conditions that may negatively impact cognitive, physical, emotional, and social development.

Moreover, many assessment tools cover a wide range of ages, such as Pedi-EAT, BAMBI, BPFAS, and many more, with an age range as wide as 6 months–7 years, 3–11 years, and 9 months–18 years respectively. However, the critical age range at which infants could develop eating problems is between 6 and 18 months, when they are finally exposed to many different foods, with different textures and tastes. This is the age at which infants develop the foundation of their oral motor skills and solid food consumption.

## 4. Discussion

Apparently, there is not yet a universally accepted definition of the term “feeding problem”. A number of terms are generally used in the research and have varied over the decades. The International Classification of Diseases (10th revision) defined children with extreme selectivity and food refusal in the presence of adequate food supply, absence of organic diseases, and being under the care of a competent caregiver as the criteria for “feeding disorder” [14]. One paper mentioned how classification is difficult due to the different terms used by researchers. Other terms include: “food refusal”, “selective eating”, “food selectivity”, “picky eating”, “fussy eating”, and “dietary restriction”. These terms should not be confused with “Avoidant/Restrictive Food Intake Disorder (ARFID)”, a diagnostic term listed in the statistical Manual for Mental Disorders V (DSM V), defining eating disorder [29]. A study by Garg et al. even uses both terms, “feeding problems” and “feeding difficulties”, in the same article, and did not differentiate between the meaning of both terms [20].

It is important to note that some screening tools have not yet been tested in TDC, but focused solely on specific clinical conditions in children, such as ASD, DS, and other neurological conditions [24,26,37]. Another condition with available assessment tools for feeding problems is Type 1 Diabetes in children, using DEPS-R (Diabetes Eating Problem Survey-Revised) [22]. This trend may be caused by previous findings regarding feeding problems commonly found in children with autism rather than other developmental disability groups, namely limited variety of food intake (picky eater), food refusal (texture, color, appearance, smell), and aggressivity during mealtime [18,26,33]. However, feeding problems are not encountered exclusively in children with health problems. For instance, problems with regulation of internal states, sensory integration and quality of caregiving and behavioral mismanagement may play important roles in the development of feeding problems in early childhood. A study in Polish using MCH-FS revealed that the most frequent mealtime behaviors in parents were walking behind the child while feeding or distracting the child with toys and/or television to ensure that children eat properly eat. Children’s refusal to eat, prolonged mealtimes, and forcing the child to eat and drink were apparently common behaviors in this study [41].

The comprehensiveness of an assessment tool depends on the number of feeding domains and the items that describe each domain. Multiple domains are necessary when evaluating feeding difficulties. However, due to the complex underlying factors regarding feeding issues, it can be challenging to identify which observable behaviors (i.e., which domains) should be included. The oral and sensory–motor domains were included in all assessment tools. It was notable that BPFAS, MCH-FS and Pedi-EAT measured these feeding domains [29,40,49].

When assessing child feeding problems, the validity of assessment tools refers to how accurately their scores reflect children’s problems. In this review, we found that MCH-FS showed excellent construct validity and reliability among Canadian samples. It has been validated and tested for reliability in French, and also translated and published in the Netherlands and Thailand [13]. Although it has some disadvantages, Pedi-EAT has been proven to have an excellent test–retest reliability, as well as internal consistency in healthy children in both its English and Arabic version [11,45]. BAMBI currently has four versions in different languages: English, Brazilian Portuguese, Italian, and Turkish [18,26,45]. However, Gal et al. had different opinions regarding BAMBI, claiming that BAMBI was not designed as thoroughly and as comprehensively as it claimed to be, due to lack of assessment regarding sameness rituals, compulsive eating behavior, and excessive eating, which commonly occurs in the ASD population [35]. DEPS-R as a tool has been tested for validity and reliability in English, Spanish, and Turkish [22].

What was universally agreed was that any signs regarding feeding problems should be addressed immediately, as the eating behavior patterns in the childhood period, particularly pre-school, tend to remain stable throughout the lifetime of the child [15]. Unfortunately, knowledge of feeding problems is not communicated sufficiently in society, resulting in caregivers thinking that their child’s feeding problems are a typical development in a growing child [53]. Preterm children should be assessed and monitored more closely, as Park et al. found that very preterm (<32 weeks gestational age at birth) and moderate-to-late preterm children (32–36 weeks gestational age at birth) had greater symptoms of feeding problems compared to full-term children, tested on children aged 6 months–7 years old [27]. Infantile anorexia, food neophobia, and sensory food aversion are usually manifested during the early stages of childhood, usually between 6 months and 3 years of age, and diminished as they grew older, although the severity of this problem could affect the child’s eating habits later in life, mainly related to a poorly balanced diet and reduced nutrient intake [29]. Early identification of feeding problems before avoidant behavioral patterns are established is crucial for selecting interventions that are timely and targeted in addressing the underlying issues [14]. The 6–18 month age period is a critical period for assessing a child’s transition to foods with more complex tastes and textures, as this process contribute to oral motor skill development. A delay in the development of both attributes may create a new problem, where children fail to try different tastes and textures appropriate to their age, hence contributing to maladaptive feeding behaviors and feeding problems in the later part of their childhood [54].

It is important to note that this review offers an evaluation of assessment tools for assessing feeding problems in children. Through this review, we were able to identify the gaps in recent tools related to usage, aspects/domains, and age range. Therefore, future research can develop new tools to fill these gaps in order to help mothers, practitioners, and researchers to identify feeding problems early before the establishment of nutritional and health problems. However, this scoping review has some limitations. Firstly, we did not analyze potentially useful missing articles, since we only included articles published in English. Second, the majority of the articles included in this study involved a very broad age range, making it difficult to understand age-specific issues. However, the present study enabled us to map articles on measurement tools for feeding problems, for whom the tools were designed, the purpose of the tools, and what aspects were included when developed them.

## 5. Conclusions

In this review, 23 assessment tools were validated and tested for reliability. Among these, Pedi-EAT, MCH-FS and BPFAS were the most commonly used instruments. However, it is clear that no single instrument provides comprehensive aspects regarding feeding problems in children. In addition, usage of the tools and the wide age range indicate that further research is needed to fill the gaps.

## Figures and Tables

**Figure 1 children-12-00037-f001:**
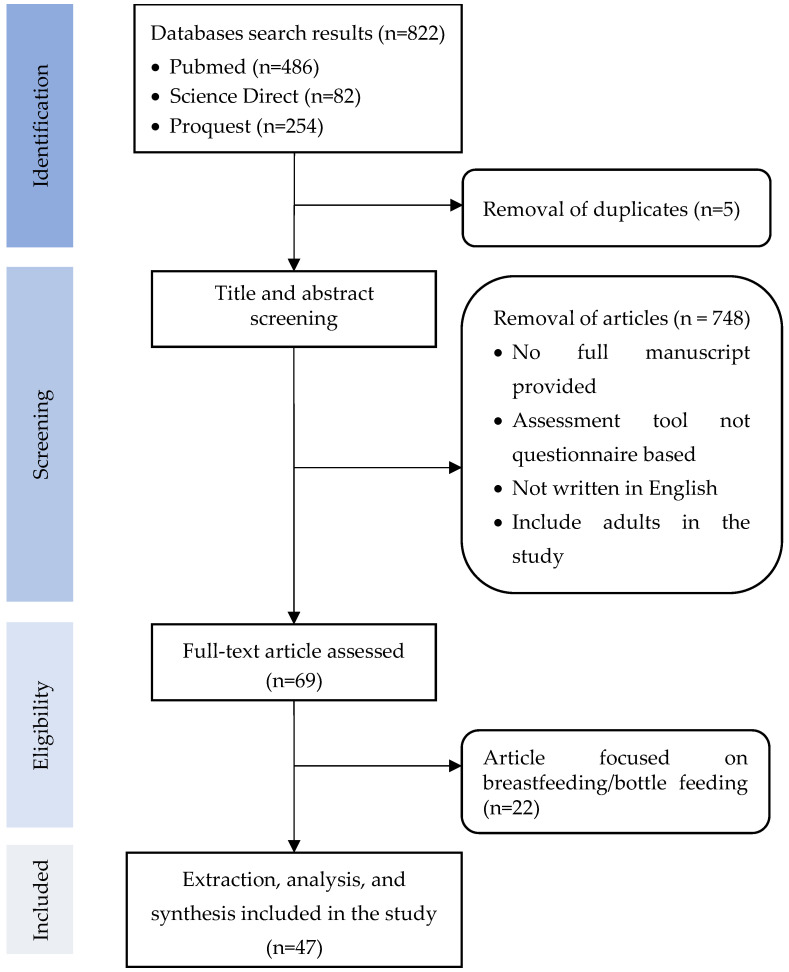
Flow diagram of article collection process.

**Table 1 children-12-00037-t001:** Summary of research on feeding problems assessment tools in children.

**No**	**Authors/Year**	**Assessment Tool**	**Population**	**Usage**	**Aspects**
** *Assessment tools for children age 6–24 months old* **	
1	Arslan, 2022 [12]	The Pediatric version of the Eating Assessment Tool-10 (PEDI EAT-10)	Children with Down Syndrome (1–3 years)	To determine the dysphagia symptom severity of children	Parent-reported;Only assess the severity of swallowing problem
2	Rogers, Ramsay and Blissett, 2018 [13]	Montreal Children’s Hospital Feeding Scale (MCHFS)	Infants (12 months)	To assess feeding problems in children	Parent-reported;Evaluate parents’ perceptions and concerns regarding mealtimes and their children’s eating, including aspects such as growth, appetite, meal duration, mealtime behavior, chewing/sucking, gagging/spitting or vomiting, food retention in the mouth, use of distractions or force to eat, and the impact of feeding on relationships.
		Child Eating Behaviour Questionnaire (CEBQ)	Children (12 months)	To measure eating behaviour traits	Assessment of food approach behaviors (such as food enjoyment, responsiveness, and desire to drink) and food avoidance behaviors (including satiety response, slow eating, and food fussiness).
		The Feeding Interaction Scale (FIS)	Children (12 months)	To assess the observed mother–infant interactions during a normal mealtime	Observation of mother–infant interactions during regular mealtimes, assessing the likelihood of reflecting how difficult or stressful mealtimes are for mothers, as well as infant food acceptance/rejection and emotional reactions during mealtime.
** *Assessment tools for children age below 5 years old* **
3	Silverman et al., 2020 [14]	Infant and Child Feeding Questionnaire (ICFQ)	Children (Birth–4 years)	To assess and describe symptoms of what might be an eating disorder on children	Caregiver-reported;Multiple choice regarding meal duration, chidren behavioral checklist regarding observed feeding problems symptoms
4	Tan et al., 2022 [15]	Child Eating Behaviour Questionnaire (CEBQ)	Children (<5 years)	To clasify the eight types of eating behaviours in children	Caregiver-reported;Assessment of food approach behaviors (such as food responsiveness, enjoyment of food, emotional overeating, and desire to drink) and food avoidant behaviors (including satiety responsiveness, slow eating, emotional undereating, and food fussiness).
5	Vries et al., 2023 [16]	Montreal Children’s Hospital Feeding Scale (MCH-FS)	Children (6–36 months) with and without clefts	To assess feeding difficulties in children with and without clefts	Assessment of parents’ perceptions and concerns regarding mealtimes and their children’s eating habits, including growth, appetite, meal duration, mealtime behavior, chewing/sucking, gagging/spitting or vomiting, food retention in the mouth, use of distractions or force to eat, and the impact of feeding on relationships.
6	Castro et al., 2024 [17](same as below)	Children’s Eating Behaviour Questionnaire for toddlers (CEBQ-T)	Children healthy (1–3 years)	To assess traits regarding appetite in toddlers	Caregiver-reported;Assessment of food approach behaviors (such as food responsiveness, enjoyment of food, emotional overeating, and desire to drink) and food avoidant behaviors (including satiety responsiveness, slow eating, emotional undereating, and food fussiness).
** *Assessment tools for children in various age ranges* **
7	Meral and Fidan, 2014 [18]	Brief Autism Mealtime Behavior Inventory (BAMBI)	Children (5–14 years)	To determine mealtime behaviors and feeding problems of children with autism	Parent-reported;Assessment of limitations in consuming a variety of foods, food refusal, disruptive mealtime behaviors, and behaviors associated with autism.
11	Marshall et al., 2015 [19]	Behavioral Pediatric Feeding Assessment Scale	Children (2–6 years)	To identify children with feeding difficulties	Parent-reported;Measurement of children’s mealtime behaviors, parents’ feelings regarding mealtime, mealtime strategies by parents
12	Garg, Williams and Satyavrat, 2015 [20]	Identification and Management of Feeding Difficulties (IMFeD)	Children (2–10 years)	To identify feeding difficulties in Indian children	Parent questionnaire and physician questionnaireMeasure limited appetite, highly selective intake, crying interferes with feeding (colic), fear of feeding
13	Bektas et al., 2016 [21]	Emotional Eating Scale Adapted for Use in Children and Adolescents (EES-C)	Children (8–17 years)	To assess the emotional eating status of children and adolescents	Three subscales denoting reasons for eating an excessive amount of food: (1) anxiety, anger, frustation, (2) Depressive symptoms and feeling unsettled
14	Altınok et al., 2017 [22]	Diabetes Eating Problem Survey-Revised(DEPS-R)	Children (9–18 years)	As a screening tool to detect disordered eating behaviors early for children with diabetes	Self-reported;Assessment of disturbed eating behaviors signs in individuals with Type 1 Diabetes
15	Lin et al., 2018 [23]	Identification and Management of Feeding Difficulties (IMFeD™)	Children (1–10 years)	To assesss parents regarding child’s feeding patterns and recommendations for intervention	Parent questionnaire and physician questionnaire;Measurement of limited appetite, highly selective intake, crying interferes with feeding (colic) and fear of feeding
16	Barton et al., 2018 [24]	STEP-CHILD (Screening tool of feeding problems applied to children)	Children age 24 months–18 yearsAutism Special needs Typically developed children	To assess feeding problems	Parent-reported;Assessment of chewing difficulties, rapid eating, food stealing, food refusal, food selectivity, and vomiting.
17	Thoyre et al., 2018 [25]	Pediatric Eating Assessment Tool (PediEAT)	Children (6 months–7 years)	To assess symptoms of feeding problems in young children	Parent-reported;Assessment of physiological symptoms, problematic mealtime behaviors, selective/restrictive eating, and oral processing issues.
18	Pados, Thoyre, and Park, 2018 [11]	Pediatric Eating Assessment Tool (PediEAT)	Children (6 months–7 years)	To assess problematic feeding symptoms in infants and young children	Parent-reported;Evaluation of physiological symptoms, disruptive mealtime behaviors, selective/restrictive eating, and oral processing issues.
19	Castro et al., 2019 [26]	Brief Autism Mealtime Behaviour Inventory (BAMBI)	Children with Autism Spectrum Disorder (ASD) (5–11 years)	To evaluate feeding problems in children with ASD	Parent-reported;Assessment of difficulties in consuming a variety of foods, food refusal, disruptive mealtime behaviors, and autism-related behaviors.
20	Park et al., 2019 [27]	Pediatric Eating Assessment Tool (PediEAT)	Children born very preterm (<32 weeks gestationalage at birth) and moderate to late preterm (32–36 weeks at birth) at 6 months to 7 years old	To assess symptoms regarding problematic feeding in infants and children	Parent-reported;Measurement of physiological symptoms, problematic mealtime behaviors, selective/restrictive eating, oral processing
21	Iron-Segev et al., 2020 [28]	Stanford Feeding Questionnaire (SFQ)	Children with avoidant/restrictive food intake disorder (ARFID)	To assess children’s eating behavior and parents’ feeding practices	Measurement of children’s eating behaviors and parental feeding problems
22	Sdravou et al., 2021 [29]	Behavioral Pediatrics Feeding Assessment Scale (BPFAS)	Children (2–7 years)	To capture feeding difficulties (nutritional and textural selectivity, food refusal, and oral motor difficulties)	Parent-reported;Assessment of nutritional and texture selectivity, food refusal, and oral motor difficulties.
23	Baqays et al., 2021 [30]	The pediatric version of the eating assessment tool (PEDI-EAT-10)	Children with cerebral palsy (18 months–18 years)	To assess swallowing dysfunction	Parent-reported;Assessment of physiological symptoms, problematic mealtime behaviors, selective/restrictive eating, and oral processing difficulties.
24	Kang et al., 2021 [31]	Behavioral Pediatrics Feeding Assessment Scale (BPFAS)	Children (1–7 years) with ASD	To measure mealtime behaviors inchildren	Assessment of the frequency of child behaviors, child behavior problems, and the frequency of parent feelings and strategies related to these issues.
25	Pados et al., 2021 [32]	Montreal Children’s Hospital-Pediatric Feeding Program (MCH-FS)	Children full-term—5 months corrected gestational age, 6–11 months, 12–23 months, and 24–48 months	To assess child feeding problems	Parent-reported;Assessment of parents’ perceptions and concerns about mealtimes and their children’s eating, including aspects, such as growth, appetite, meal duration, mealtime behavior, chewing/sucking, gagging/spitting or vomiting, food retention in the mouth, use of distractions or force to eat, and the impact of feeding on relationships.
26	Adams, Verachia and Coutts, 2022 [33]	Behavioural Paediatric Feeding Assessment Scale (BPFAS)	Children (3-<10 years) with ASD	To identify feeding difficulties, disorders and problematic mealtime behaviours in children	Measurement of child behavior frequency, child behavior problems, parent feelings and strategies frequency problems
27	Nakaoka et al., 2022 [34]	Autism Spectrum Disorder Mealtime Behavior Questionnaire (ASD-MBQ)	Children (3–18 years)	To assess mealtime behavior in Japanese children	Caregiver-reported;Assessment of problematic behaviors, such as selective eating, clumsiness/manners, attention/concentration during eating, and oral–motor function.
28	Gal et al., 2022 [35]	Autism Eating (Aut-Eat) Questionnaire (AEQ)	Children (3-<8 years)	To assess eating problems and patterns in children with ASD	Parent-reported;Assessment of demographic and developmental information related to the child’s ASD, six types of eating problems (chewing and swallowing difficulties, food avoidance, selective eating, eating rituals and sameness, overeating, and problematic mealtime behaviors), and a list of foods consumed by the child.
29	Adel et al., 2022 [36]	Pediatric eating assessment tool (Pedi-EAT-10)	Children (>6 months)	To determine children with high risk of dysphagia	Parent-reported;Assessment of physiological symptoms, mealtime behavior issues, selective/restrictive eating, and oral processing difficulties.
30	Rouphael et al., 2023 [37]	Behavior Pediatrics Feeding Assessment Scale (BPFAS)	Healthy children and ASD children (6–9 years)	To measures the risk of feeding problems	Assessment of the frequency of child behaviors and behavior problems, as well as the frequency of parent emotions and strategies used to address these issues.
		My Child Eating Habits (MCEH)	Healthy children and ASD children (6–9 years)	To compare feeding difficulties among children without autism and with autism	Evaluation of the nature of feeding issues, food preferences, food refusal, potential causes of the child’s eating habits, specific parental concerns, and parents’ willingness to pursue feeding therapy.
31	Bialek-Dratwa and Kowalski, 2023 [38]	Montreal Children’s Hospital-Pediatric Feeding Program (MCH-FS)	Children (6 months–6 years)	To assess food neophobia on children	Parent-reported;Assessment of feeding characteristics (oral motility, oral sensory issues, and appetite), mothers’ concerns about feeding, mealtime behaviors, strategies employed by mothers during feeding, and family responses during child feeding. Evaluation of children’s interest in trying new foods and their preference for specific foods.
		Food Neophobia Scale for Children (FNSC)		
32	Bialek-Dratwa and Kowalski, 2023 [38]	Food Neophobia Scale—Children (FNSC)	Children (2–7 years)	To assess a person’s food neophobia level	Assessment of feeding characteristics (oral motility, oral sensory issues, and appetite), mothers’ concerns about feeding, mealtime behaviors, strategies employed by mothers during feeding, and family responses during child feeding. Evaluation of children’s interest in trying new foods and their preference for specific foods.
33	Shabnam and Swapna, 2023 [39]	Feeding handicap index for children (FHI-C)	Children (2–10 years)	To assess feeding problems and their impact in children with developmental disabilities	Caregiver-reported;Physical domain (feeding ability), functional domain (swallowing ability, introducing new foods), emotional domain (emotions during eating)
34	Sforza et al., 2023 [40]	Montreal Children’s Hospital Feeding Scale (MCH-FS)	Children (6 months–6 years)	To identify the severity of feeding difficulties in children	Parent-reported;Assessment of feeding characteristics including oral motility, oral sensory issues, and appetite, as well as mothers’ concerns about feeding, mealtime behaviors, strategies employed by mothers during feeding, and family reactions during child feeding.
35	Bialek-Dratwa and Kowalski, 2023 [41]	Montreal Children’s Hospital Feeding Scale (MCH-FS)	Children (2–7 years)	To assess oral motility, sensory, and appetite	Assessment of feeding characteristics including oral motility, oral sensory issues, and appetite, as well as mothers’ concerns about feeding, mealtime behaviors, strategies employed by mothers during feeding, and family reactions during child feeding.
36	Abild et al., 2023 [42]	Diabetes Eating Problem Survey Revised (DEPS-R)	Adolescents (11–18 years) with type 1 diabetes	To detect disordered eating in adolescent with diabetes	Self-reported;Assessment of disturbed eating behaviors signs in individuals with Type 1 Diabetes
37	Bykova, Fank, and Girolami, 2023 [43]	Eating and Drinking Ability Classification System (EDACS)	Children (2–6 years) with celebral palsy	To classify eating and drinking efficiency and safety in children with cerebral palsy	Evaluation of dependency during mealtime, classify eating, drinking, and swallowing ability
38	Alibrandi et al., 2023 [44]	Brief Autism Mealtime Behavior Inventory (BAMBI)	Children with ASD (3–11 years)	To assess behaviour problems and evaluate feeding problems in children with ASD	Parent-reported;Assessment of difficulties in consuming a variety of foods, food refusal, disruptive mealtime behaviors, and behaviors associated with autism.
39	Lamboglia et al., 2023 [45]	Brief Autism Mealtime Behavior Inventory (BAMBI)	Children with ASD (3–11 years)	To assess mealtime behaviors specific to children with ASD	Parent-reported;Evaluation of difficulties in consuming various foods, food refusal, disruptive mealtime behaviors, and behaviors associated with autism.
40	Bresciani et al., 2023 [46]	Brief Autism Mealtime Behavior Inventory (BAMBI)	Children with ASD (1–10 years)	To capture mealtime behaviors specific to children with ASD	Parent-reported;Evaluation of difficulties in consuming a range of foods, food refusal, disruptive mealtime behaviors, and autism-related behaviors.
		Behavior Pediatric Feeding Assessment Scale (BPFAS)	Children	To assess eating behavior	Evaluation of frequency of child behaviors and behavior problems, as well as the frequency of parental emotions and strategies used to address these issues.
41	Chebar-Lozinsky et al., 2024 [47]	Questionnaire used by Wright et al. (2007) for the UK Millennium Study	Children (4 weeks–16 years)	To assess eating problems, eating and feeding behavior, food and food type preferences, and drinking pattern	Assessment of daily meal refusal, prolonged mealtimes (defined as lasting over 30 min), gagging on textured foods, poor appetite, dysphagia (indicated by visible signs of difficulty swallowing), and challenges with sucking (whether from breast or bottle).
42	Wang et al., 2024 [48]	The Pediatric Eating Assessment Tool-10 (Pedi-EAT-10)	Children (3–11 years)	To assess swallowing problems and dysphagia	Parent-reported;Assessment of physiological symptoms, disruptive mealtime behaviors, selective/restrictive eating, and oral processing difficulties.
43	Brunner et al., 2024 [49]	Pediatric Eating Assessment Tool (PediEAT)	Children (6 months–7 years)	To identify symptoms of problematic feeding in children	Parent-reported;Assessment of physiological issues, challenging mealtime behaviors, selective or restrictive eating, and difficulties with oral processing.
44	Harrington et al., 2024 [50]	Diabetes Eating Problems Survey-Revised (DEPS-R)	Children (11–14 years)	Parent-reported screening to identity disordered eating in children and young people with Type 1 Diabetes	Parent self-reported;Measurment of disturbed eating behaviors signs in individuals with Type 1 Diabetes
45	Bialek-Dratwa and Kowalski, 2024 [51]	Food Neophobia Scale Children (FNSC)	Children (2–7 years)	To assess a person’s level of food neophobia and propensity to try unfamiliar foods	Parent-reported;Assessment of feeding characteristics (oral motility, oral sensory and appetite), mothers’ concerns about feeding, mealtime behavior, strategies used by mothers during feeding, and family response during child feeding
46	Xie et al., 2024 [52]	Child Food Neophobia Scale (CFNS)	Children (2–7 years)	To assess children’s fear of new food	Parent-reported;Assessment of feeding characteristics (oral motility, oral sensory and appetite), mothers’ concerns about feeding, mealtime behavior, strategies used by mothers during feeding, and family response during child feeding
47	Castro et al., 2024 [17]	Aut-Eat Questionnaire	Children with ASD	To capture eating problems and patterns in children with ASD	
		Behavioral Pediatrics Feeding Assessment Scale (BPFAS)	Children with ASD	To assess various problematic and desirable eating behaviors and their behavior during meals	
		Children’s Eating Behavior Inventory (CEBI)	Children with ASD and healthy	To evaluate eating problems and eating behavior in children based on the child, parents, and family factors	
		Child Eating Behavior Questionnaire (CEBQ)	Children with ASD and healthy	To assess food approach behaviors	
		Children’s Eating Behaviour Questionnaire for toddlers (CEBQ-T)	Children healthy (1–3 years)	To assess traits regarding appetite in toddlers	
		Mealtime Behavior Questionnaire (MBQ)	Children with ASD and healthy (2–6 years)	To assess the types of eating difficulties in children	
		Montreal Children’s Hospital Feeding Scale (MCH-FS)	Children with ASD and healthy	To identify the severity of feeding difficulties in children	
		Pedi-EAT	Healthy children	To evaluate problematic eating behaviors in children	
		Screening Tool of Feeding Problems applied to children (STEP-CHILD)	Children with ASD	To identify eating problems in both the child and the parents

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
