# Peer review of "Feeding Problems Assessment Tools in Children: A Scoping Review"

_children, 2024, doi:10.3390/children12010037_

Round 1

Reviewer 1 Report

Comments and Suggestions for Authors

This is an interesting scoping review of feeding problem assessment tools in children. However, if possible, it could be improved.

1.    The results section is mostly the table of 47 reviewed articles with 21 tools. However, there appears to be much information on the tools that was written in the discussion section but is not in the results. Please move that information to the results section.  

2.    The conclusion section should build on what was written in the article. The paper concludes that “there are several available tools that were validated and tested for reliability” and that the tools have problems of “internal consistency, poor reliability, and failure to include patients’ input in the development of the questionnaire, makes some screening tools need further study and development.” The paper gives information on the internal consistency, validity, and reliability of some of the tools but not for others. In addition, I couldn’t find any discussion of patient input. If possible, add more information and, when applicable, move it to the results section.

3.    The conclusion also states, “The wide age range of several tools should be reassessed due to different development stages of those age range. There should be a tool to assess the period between 6 – 24 months old as it was the critical stage where children were finally exposed to many different foods.” Again, this should have been discussed earlier.

Author Response

Comment:

  1. The results section is mostly the table of 47 reviewed articles with 21 tools. However, there appears to be much information on the tools that was written in the discussion section but is not in the results. Please move that information to the results section.  

Response:

Thank you for your observation, we rearrange some parts of the Discussion section into Result section to accommodate your suggestion which highlighted with yellow colour. We also added additional descriptions of the assessment tool in Results section and relevant descriptions in Discussion section which highlighted with blue colour.

Comment:

  1. The conclusion section should build on what was written in the article. The paper concludes that “there are several available tools that were validated and tested for reliability” and that the tools have problems of “internal consistency, poor reliability, and failure to include patients’ input in the development of the questionnaire, makes some screening tools need further study and development.” The paper gives information on the internal consistency, validity, and reliability of some of the tools but not for others. In addition, I couldn’t find any discussion of patient input. If possible, add more information and, when applicable, move it to the results section.

Response:

We improved the statement in the Conclusion which can be seen in line 330-334. In addition to support this conclusion, we also improved relevant parts in the Result section, such as: the tools that commonly used in line number 168-172; description regarding comprehensiveness of the tool in line 196-199 (Result section) and in line 275-281 (Discussion section), while related with age range, we added some information in Result and Discussion in line 233-238 and 304-315, respectively.

Comment:

  1. The conclusion also states, “The wide age range of several tools should be reassessed due to different development stages of those age range. There should be a tool to assess the period between 6 – 24 months old as it was the critical stage where children were finally exposed to many different foods.” Again, this should have been discussed earlier.

Response:

Your comments have been accommodated in the Results and Discussion section to be aligned with the Conclusions section which can be found in line 233-238 and 304-315 respectively.

Reviewer 2 Report

Comments and Suggestions for Authors

Comments attached

Author Response

Comment:

  1. Introduction reads mostly well and provides a succinct background to the topic. When referring to feeding problems the author can consider adding information on parenting styles, specifying each definition would help to place the reader in the context of this paper

Response:

Thank you for your input, however in this scoping review on feeding problems assessment tools, we feel that the focus is not on parenting style but role of caregivers. However, this aspect has been added in the Introduction section in line 58-59.

Comment:

  1. Consider introducing that the terms "feeding problems" or "feeding difficulties" are interchangeable or not earlier in the article to avoid confusion for the reader.

Response:

Your concern has been addressed in the Introduction section in line 43-47 and highlighted in the Discussion section in line 246-258

Comment:

  1. Why did you choose to include prevalence data from Poland and India? Would it be beneficial to include data from other regions for a broader perspective?

Response:

Thank you for your concern. We decided to remove data from Poland and India  and change them using  references number 1 and 2.

Comment:

  1. Could the author elaborate on how this scoping review is expected to benefit academics, practitioners, and parents?

Response:

This concern has been included in the Discussion section in line 316-321.

Comment:

  1. Are the research questions broad enough to cover all necessary aspects of feeding problem assessment tools, or could they be refined for greater focus?

Response:

The aim of this paper is to map the existing assessment tools on feeding problems /difficulties. We believe, the research questions applied for this scoping review is suffice to achieve this aim.

Comment:

  1. Font is different in different sections

Response:

The fonts are in uniform size in the revised manuscript

Comment:

  1. There are some abbreviations in the abstract, please write the full names when first mentioned.

Response:

All abbreviations have been written in full names when first mentioned

Comment:

  1. The data sources and methods for study inclusion and time period of inclusion are well described and supported

Response:

Thank you for your comment

Comment:

  1. Flow diagram is well constructed

Response:

Thank you for your comment

Comment:

  1. The use of bullet points in the table disrupts the flow and clarity of the content; consider removing them for a more cohesive presentation. Please improve the format by justifying the contents.

Response:

Thank you for your suggestion, the bullet points have been removed from the table.

Comment:

  1. In the discussion, the author has highlighted the limitations of Pedi-EAT and BAMBI. Could you specify if there are any emerging tools or methods that address these shortcomings?

Response:

Thank you for your comment, however the purpose of this scoping review is not to recommend improvements for the weaknesses found in each tool, but rather to assess the gaps in the existing tools related to availability, implementation, and implications.

Comment:

  1. Conclusion should be presented objectively and highlight the most appropriate findings that address the research question. It is also important to state whether mixed results are present and whether there is a need for further study on a specific theme.

Response:

Your concern has been addressed by improving  the statement in the Conclusion section which can be seen in line 330-334. In addition to support this conclusion, we also improved relevant parts in the Result section, such as: the tools that commonly used in line number 168-172; description regarding comprehensiveness of the tool in line 196-199 (Result section) and in line 275-281 (Discussion section), while related with age range, we added some information in Result and Discussion in line 233-238 and 304-315, respectively.

Comment:

  1. Nevertheless, the review may contribute to underlying the need of better studies on this topic and it may be useful to obtain information on child feeding problem

Response:

Thank you for your comment

Comment:

  1. Please correct references as per guidelines (e.g Ref#18, 35, 44, 50, 60)

Response:

The references have been corrected based on guidelines.

Round 2

Reviewer 1 Report

Comments and Suggestions for Authors

Thanks for responding your careful edits to the manuscript.

Reviewer 2 Report

Comments and Suggestions for Authors

no further comments